# D-CHOPT: DISCOVERING CLOSED-FORM HIGH-DIMENSIONAL ODEs FROM PARTIAL OBSERVED TRAJECTORIES

## ABSTRACT

Machine learning algorithms have become a new paradigm for automatically discovering closed-form ordinary differential equations (ODEs) from observed trajectories. Although significant breakthroughs have been made in this field, such as symbolic regression and sparse identification of nonlinear dynamics (SINDy), existing approaches primarily perform well for low-dimensional ODEs. This limitation arises due to the lack of understanding of observability limitations in partially observed trajectories, and the additional challenges introduced by complex topological properties. In this work, we propose a method for discovering closed-form high-dimensional ODEs from partially observed trajectories, called D-CHOPT, which advances ODE discovery methods beyond the natural limitations of high-dimensional ODEs. D-CHOPT uses an invertible neural network as the backbone to find the optimal solution within the diffeomorphic equivariant group of the reconstructed dynamical systems, while preserving topological properties and integrating a variable selection method. We provide a formal analysis of observability and the learning limitations of partial trajectories, and explain the enhancements in a manner consistent with the theoretical results. In experiments, D-CHOPT successfully discovered the governing equations for a wide range of dynamical systems, both low and high dimensional.

## 1 INTRODUCTION

An ordinary differential equation (ODE) is a significant research object in scientific fields, where the solution is a high-dimensional curve that evolves over time. The mathematical expression of ODE is the one that builds the bridge between the continuous state $\mathbf{x}(t)$ of the system and its time derivative $\dot{\mathbf{x}}(t)$ via a function $f$ as $\dot{\mathbf{x}}(t) = f(\mathbf{x}(t), t)$. Closed-form of the ODE if f has a concise and analytic expression. Closed-form plays an important role both in engineering applications and scientific research because it provides explicit information that can be used to explain how factors interact and influence the evolution of the entire system (Shakeel et al. (2018); Kacprzyk et al. (2024)). Also, the concrete expression of the ODE facilitates the analysis of the whole system, e.g. (Banerjee (2021)), and helps to reveal the physical rules.

Identifying and discovering the closed-form of ODEs based on a human expert is a time-consuming process . However, automatically discovering closed-form of ODEs has gained interest for the machine learning community and has become one of the important topics of AI for science (Wang et al. (2023)). Several methods have been proposed for this task. Sparse Identification of Nonlinear Dynamics (SINDy) (Brunton et al. (2016)) is an established approach to discover a closed-form function of an ODE model. The central idea is to use sparse linear regression to uncover parsimonious governing equations from a dictionary of basis functions constructed by data, where the sparsity is promoted by pruning out redundant terms based on certain specified thresholds. In the past few years, the SINDy framework has been further improved in various aspects to address these challenges (Hirsh et al. (2022)), e.g., enhancing the library or using deep learning for denoising and derivative computation by fitting the noisy data in a decoupled or coupled manner, for instance, filter techniques such as Savitzky-Golay filter can both reduce noise and compute numerical derivatives (Egan et al. (2024)). For uncertainty quantification, the dictionary-based equation discovery algorithms have been recently extended to Bayesian settings, based on the idea of sparse Bayesian

learning pioneered by Tipping and co-workers (Fasel et al. (2022), Champneys & Rogers (2025)). Another popular method is the symbolic regression method, which estimates the time derivatives as the label and employs the symbolic regression method combined with a certain optimization method to search for the optimal function and possible functional form (Brunton et al. (2016)).

However, both methods are not applicable to partially observed ODEs, because we typically do *not* observe the full state $\mathbf{x}_t$. Consequently, we only obtain ODE expressions corresponding to the dimensions of the low-dimensional observations, rather than expressions for the full set of state variables. We refer to this problem as the *attractor-dimension mismatch*.

A simple way to resolve this mismatch is to recover the full state $\mathbf{x}_t$ from partially observed data. The delay-coordinate mapping technique bridges this gap, and under certain conditions given by Takens' embedding theorem (Takens (2006)), which yields an attractor that is diffeomorphic to the hidden full-state attractor. Based on this embedding, neural networks can be trained to approximate the inverse mapping, thereby transforming the reconstructed dynamics back to the original system. Once this is achieved, closed-form governing equations can be discovered using SINDy (Bakarji et al. (2023), Champion et al. (2019)).

Although delay-coordinate embedding for high-dimensional hidden attractor embedding has a rich history in data science, especially in time series prediction, causal discovery (Sugihara et al. (2012)), and data-driven modeling (Kim et al. (1999)), due to an insufficient understanding of the theorem and consideration of the properties of ODEs, i.e. the observability of variables which naturally limits the learning ability of the learning framework, the current learning framework has not organically integrated the neural network within the corresponding theoretical foundations and lacks sufficient constraints, merely allowing the network to pick a solution from the solution space and limit its utility for low-dimensional ODEs.

In this work, we develop the Discovery of Closed-form High-dimensional ODE from Partially Observed Trajectories framework (D-CHOPT), which improves the learning accuracy for SINDy-AE and extends it beyond the limitation of low-dimensional settings. The key insight behind D-CHOPT lies in the observability of variables (Letellier & Aguirre (2002)), and the local diffeomorphism property of delay-coordinate mapping (Cross & Gilmore (2010b)), which establish direct links between the partial observation and the ODE $f$. This approach overcomes limitations of partial trajectories and provides a rigorous and practical solution for the discovery of the true (but unknown) ODE using partially observed trajectories. We establish a framework to address this limitation and propose an algorithm to select the optimal measurements among multiple partially observed trajectories. Under this framework, we propose the topology preserving invertible flow neural networks. We demonstrate via extensive experiments that D-CHOPT can uncover the governing equations for a wide range of dynamical systems while being more accurate and successfully discovering higher-dimensional systems than the alternative methods. Finally, D-CHOPT highlights the importance of topology and observability in partially observed trajectories for the representation learning community when dealing with time series data observed from dynamic systems.

## 2 BACKGROUND AND PROBLEM SETTING

Dataset consisting of trajectories for partial ODE discovery, especially under the partially observed trajectories with delay-coordinate mapping, involves many decisions, e.g., how to choose the optimal embedding parameters? When and for how long to take measurements? (we discuss these questions in Appendix B.) In this work, we aim to solve the most important question under the partially observed trajectories case: how to select the optimal measurement for Takens' embedding theorem (when the observed trajectories are over two) and how to discover the latent dynamic system under a scenario with incomplete information. We assume that the dataset is *given* for which the variables can be modeled by a system of first-order autonomous ODEs Ginoux (2009) because higher-order ODEs and non-autonomous ODEs can be translated into this form by using D'Alembert transformation or adding additional variables, separately. A dynamical system is defined as

$$\frac{d\mathbf{x}(t)}{dt} = \mathbf{f}(\mathbf{x}(t); \mu), \tag{1}$$

where $\mathbf{x} \in \mathcal{M}$ is a $m$-dimensional time-dependent state vector defined on a smooth compact sub-manifold of $\mathbb{R}^J$, with $m > 0$ and $t > 0$. $\mathbf{f}$ is a smooth and nonlinear function and $\mu$ is a vector of

parameters of the system. Further, in the case with noise, the noisy states $\mathbf{y}(t)$ are given by

$$\mathbf{y}(t) = \mathbf{x}(t) + \epsilon(t) \tag{2}$$

where $\epsilon(t)$ denotes the noise process. Our goal is to learn an approximate dynamic system from the noisy measurements. Usually, in order to observe the values of a dynamic system, we need some observers, i.e. a function $h : \mathbb{R}^m \mapsto \mathbb{R}^d$ to access the measurements of the dynamic system.

In this work, we consider the measurement function $h$ of the ODEs as the coordinate projection function at discrete times, that is, $h(\mathbf{x}(t)) = \mathbf{y}(t)$. If the observation data for the full state $\mathbf{x}(t)$ are available, that is $h(\mathbf{x}(t)) \in \mathbb{R}^m$, i.e. $(d = m)$, the approximation of $\mathbf{f}$ and derivative $\dot{\mathbf{f}}$ can be inferred using various existing methods (Qian et al. (2022)). However, in many applications, only partial measurements are available, which means the dimension of $h(\mathbf{x}(t))$ $d$ is less than $m$. In this situation, direct approaches like SINDy, symbolic regression, or other universal models that rely on the full-state information do not generalize well.

**Delay Embedding.** In order to address the issue caused by partial observation, several embedding techniques have been proposed to enrich the information (Sauer et al. (1991)). One of the popular techniques is the delay-coordinate mapping based on Takens' embedding theorem. The delay-coordinate mapping $\phi(t; n, \tau) = [y(t), y(t+\tau), y(t+2\tau), ..., y(t+(m-1)\tau)]$, where $n$ and $\tau$ are the embedding dimension and the time delay, and $y(t)$ is a single coordinate from $\mathbf{y}(t)$. Here, we assume the embedding dimension of the reconstructed dynamic system is the same as the original one. We give a further discussion about the parameter selection of delay-coordinate mapping in Appendix B. The reconstructed system can be assembled into a Hankel matrix (Hirsh et al. (2021)):

$$\mathbf{H} = \begin{bmatrix} y(t) & y(t+\tau) & \cdots & y(t+q\tau) \\ y(t+\tau) & y(t+2\tau) & \cdots & y(t+(q+1)\tau) \\ \vdots & \vdots & \ddots & \vdots \\ y(t+(m-1)\tau) & y(t+m\tau) & \cdots & y(t+(m+q-1)\tau) \end{bmatrix} := [\mathbf{h}_1, \mathbf{h}_2, \ldots, \mathbf{h}_q], \tag{3}$$

where $m$ denotes the embedding dimension, $q$ is the number of discrete samples, and $\tau$ is the time delay. Takens' embedding theorem provides theoretical conditions for when time-delay embedding results in an attractor that is diffeomorphic to the original system, which means the embedding mapping $\phi$ is differentiable and invertible *almost everywhere*. Consequently, a natural idea is approximating the inverse of the delay-coordinate mapping $\psi$ using a Neural Network and then we can recover the original dynamic system.

**Subtleties of Takens' Embedding Theorem for closed-form discovery.** Unfortunately, there is quite a significant misunderstanding of Takens' theorem in previous work. The existence of embedding is guaranteed by the Whitney Embedding theorem (Whitney (1944)), which states that any smooth real $m$-dimensional manifold can be smoothly embedded in the space with dimension larger than $2m$ without any hint about how to find such embedding. And Takens' embedding theorem provides a concrete construction that under certain conditions (generic choice of $\mathbf{f}$ and $\mathbf{h}$), the delay-coordinate mapping is an embedding.

Ideally, an embedding of an $m$-dimensional dynamic system is $m$-dimensional, which aligns with our goal of closed-form discovery since we want to recover the function forms of the original system rather than learning the function forms of its diffeomorphic system. However, in reality, achieving this goal is challenging. For example, in the case of the Lorenz63 system, it has been proven that the minimum dimension required for embedding this system is four, and no three-dimensional embedding exists for the Lorenz63 system (Cross & Gilmore (2010a)). And the same issue exists for the delay-coordinate mapping, which we use to reconstruct the dynamical attractor manifold.

In order to investigate this problem, we start from the continuous form of the delay-coordinate mapping, i.e. the differential mapping, which has a better analytic property. And the approximation error is guaranteed by the following Theorem 5. We observe that the differential mapping, is **not** an (global) embedding, as the Jacobian matrix degenerates in certain regions of the attractor, which we call it as the *singular manifold* of the ODE system, denoted as $\mathcal{M}_s$. The most intuitive manifestation of this is that, for variables with low observability, the trajectory of the reconstructed dynamic system obtained through delay coordinate mapping no longer remains smooth. In regions near the singular manifold, it can exhibit sharp fluctuations or even self-intersections, as illustrated in Fig. 1. For explanation and fully uncover $\mathbf{f}$, we start from the following definition of observability for $\{\mathbf{f}, \mathbf{h}\}$.

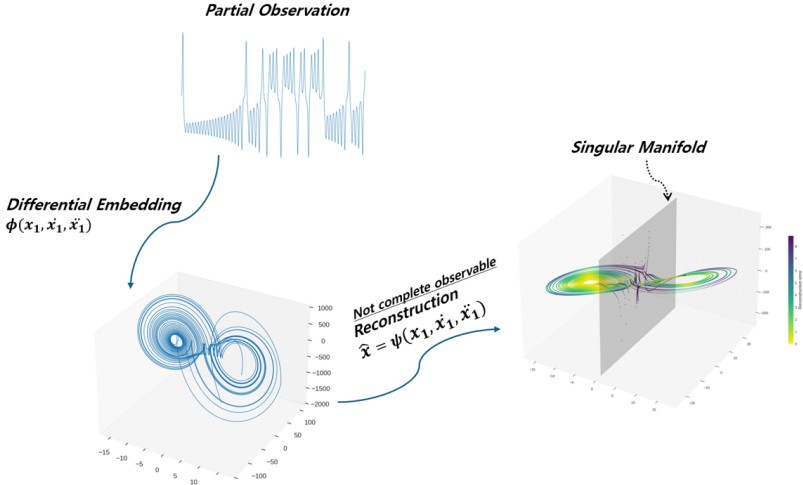

Figure 1: Reconstruction of the original coordinates of the Lorenz system from the differential embedding under the partial observation $h(\mathbf{x}) = x_1$. The system is unobservable at the intersection of the singular manifold and the Lorenz attractor ($x_1 = 0$). As a result, the reconstructed dynamic system shows a large error closer to this region.

**Definition 1.** The observable space $\mathcal{O}(\mathbf{x})$ of ODE system is the linear space of functions over the field $\mathbb{R}$ spanned by the following functions (Sendiña-Nadal & Letellier (2022)):

$$\mathcal{L}_{\mathbf{f}}^v \mathbf{h}(\mathbf{x}) := [\mathcal{L}_f^v h_1(x)...\mathcal{L}_f^v h_q(\mathbf{x})]^T, 0 \leq v \leq s, \quad 1 \leq j \leq q \quad (4)$$

where $\mathcal{L}_f^v h_j(x)$ denotes the $v$-th Lie derivative of the $j$-th component of $\mathbf{h}(\mathbf{x})$ along $\mathbf{f}$, and $s$ is the smallest integer such that $\nabla \mathcal{L}_{\mathbf{f}}^k \mathbf{h}(\mathbf{x})$ belongs to the span formed by the ODE for all $k > s$.

Due to the nonlinearity, the definition of observability is local, depending on the point $\mathbf{x}$ in the state space where the attractor manifold resides. To quantify observability, we present the following theorem and proposition (Montanari et al. (2022)).

**Theorem 2.** *If the ODE system in Eq. (1) together with observable function $\mathbf{h}$ is locally observable at $\mathbf{x}_0$, then there exists a neighborhood of $\mathbf{x}_0$ such that*

$$\dim\{\nabla \mathcal{O}(\mathbf{x}) = \mathrm{m}\}. \quad (5)$$

**Proposition 3.** *Given some measurement $\mathbf{h}(\mathbf{x})$, a differential embedding between $\mathcal{O}(\mathbf{x})$ and the original space can be constructed using higher-order derivatives of $\mathbf{h}(\mathbf{x})$ as coordinates:*

$$\Phi(\mathbf{x}) = \begin{bmatrix} \mathbf{y} \\ \dot{\mathbf{y}} \\ \vdots \\ \mathbf{y}^{(v)} \end{bmatrix} = \begin{bmatrix} \mathbf{h}(\mathbf{x}) \\ \frac{d\mathbf{h}(\mathbf{x})}{dt} \\ \vdots \\ \frac{d^v \mathbf{h}(\mathbf{x})}{dt^v} \end{bmatrix} = \begin{bmatrix} \mathcal{L}_{\mathbf{f}}^0 \mathbf{h}(\mathbf{x})^\top \\ \mathcal{L}_{\mathbf{f}}^1 \mathbf{h}(\mathbf{x})^\top \\ \vdots \\ \mathcal{L}_{\mathbf{f}}^v \mathbf{h}(\mathbf{x})^\top \end{bmatrix}, \quad (6)$$

*If the pair $\{\mathbf{f}, \mathbf{h}\}$ is observable, then such a differential embedding map is left invertible. Following the inverse function theorem, $\Phi$ is invertible if its Jacobian matrix has full rank.*

In this paper, we only focus on the case where the dimension of $\mathbf{h}(\mathbf{x})$ is one. Here, we reveal the relationship between the delay-embedding mapping and the differential embedding mapping.

**Remark 4.** In the limit of the discretization step $\Delta t \to 0$, the column space of the Hankel matrix obtained by the delay-embedding mapping is linearly isomorphic to the differential mapping up to $m$-dimensional higher-order infinitesimals.

**Theorem 5.** *(Beckermann & Townsend (2019)) Let $\mathbf{H}_n \in \mathbb{R}^{n \times n}$ be a positive definite Hankel matrix, with singular values $\sigma_1 \geq ... \geq \sigma_n$. Then $\sigma_j \leq C\rho^{-j/\log n}\sigma_1$ for some constants $C$ and $\rho$ for $i = 1, ..., n$.*

A direct result is that the observable space $\mathcal{O}(\mathbf{x})$ of the reconstructed dynamic system for both differential embedding and delay-coordinate mapping is the same. And the Theorem 5 reveals that the differential mapping can be approximated up to an accuracy of $\epsilon||\mathbf{H}||_2$ by a rank of $\mathcal{O}(\log n \log 1/\epsilon)$ matrix which provided the error upper bound for our approximation. We can analyze the dynamic system reconstructed from the delay-coordinate mapping by analyzing the differential mapping. For example, we can consider the differential embedding $\phi = (x_1, \dot{x}_1, \ddot{x}_1)$ for the reconstruction of the Lorenz attractor, which we can observe in Figure 1.

## 3 MODEL AND ALGORITHM

In this paper, we first propose a variable selection algorithm to solve the problem we proposed in the last section. Moreover, based on the variable with optimal observability, we propose a learning framework called D-CHOPT, which is an end-to-end interpretable learning framework for *system identification* that provides a closed form for dynamic systems with partially observed trajectories. In the sequel, we first recall some preliminaries and notations on system identification. Then we continue with the model architecture and techniques.

### 3.1 SYSTEM IDENTIFICATION

System identification aims to uncover a closed-form representation of the unknown ordinary differential equations. Existing methods can be broadly categorized into two groups: sparse regression and symbolic regression (Chiuso & Pillonetto (2019)). Both approaches use regression techniques combined with optimization to select the most probable basis functions from a set of candidates. The key difference lies in their assumptions: sparse regression assumes that the true ODEs can be expressed as a linear combination of these candidate basis functions, whereas symbolic regression lifts this linearity constraint and allows for discovering closed-form expressions in a more flexible, potentially nonlinear form. The main challenge in system identification is the time derivative, which is often unobserved due to difficulties in direct measurement, especially in cases with infrequent or noisy observations. Moreover, the symbolic regression suffers more because of the larger search space. Most existing work addresses this issue by employing robust derivative estimation methods (Rosafalco et al. (2025)) or bypassing the unobserved time derivatives through a variational formulation of ODEs (Qian et al. (2022)).

### 3.2 LEARNING PARTIAL OBSERVED DYNAMICS

Partial observed trajectories are the norm for practical applications and become a key problem for discovering closed-form dynamics, and several efforts have been made in this field, such as a neural operator for solving the attractor dimension mismatch or a delay-coordinate mapping as the bridge for the gap Young & Graham (2023). It utilize the low-dimensional observations for reconstructing the attractor with the same dimension as the original ones and then applying regression technique (sparse regression/symbolic regression) to obtain the closed form of the original system, for example, the Neural-ODE based Neural Delay Differential Equations (Chen et al. (2018)) which models the latent original dynamic system based on the time-delay reconstruction of the observed system. Importantly, these types of neural methods do not provide a concise closed-form expression for the latent dynamics, and moreover, these types of models do not deeply investigate the nature properties of dynamic systems, for example, the observability of dynamic variables. Using an inappropriate set of observed trajectories for reconstructing the original attractor may cause failure of the system with complex coupling, e.g., chaotic systems, which act as the starting point for our work.

## 4 METHOD AND ALGORITHMS

### 4.1 VARIABLE SELECTION ALGORITHM

A pressing question is how to select the optimal measurement to achieve the goal for embedding if we have multiple measurements, for example, $h(\mathbf{x}) = x_1$, or $h(\mathbf{x}) = x_2$, since observability greatly influences the quality of embedding.

| System | observability order | Order of Singular manifold | Percentage of the first component |
|---|---|---|---|
| Rössler | $y \approx x \triangleright z$ | $dim(\mathcal{M}_y) = 0$ | **79.30366** $\pm$ 1.40010 |
| | | $dim(\mathcal{M}_x) = 1$ | 79.14888 $\pm$ 1.46845 |
| | | $dim(\mathcal{M}_z) = 2$ | 47.82087 $\pm$ 0.66325 |
| Lorenz'84 | $x \triangleright y \approx z$ | $dim(\mathcal{M}_x) = 1$ | **78.65237** $\pm$ 1.35443 |
| | | $dim(\mathcal{M}_y) = 3$ | 71.81645 $\pm$ 1.40565 |
| | | $dim(\mathcal{M}_z) = 3$ | 70.03762 $\pm$ 0.94000 |
| Lorenz | $x \triangleright y \triangleright z$ | $dim(\mathcal{M}_x) = 1$ | **54.38915** $\pm$ 0.74183 |
| | | $dim(\mathcal{M}_y) = 3$ | 53.81355 $\pm$ 0.71960 |
| | | $dim(\mathcal{M}_z) = 2$ | 52.67460 $\pm$ 0.57763 |

Table 1: Results for the benchmark models from variable selection algorithm.

Theoretically, we can calculate the differential embedding of the dynamical variable to evaluate the observability of the reconstructed space through the degree of the singular manifold. For example, the algebraic order of the singular manifold of the Rössler system with $h(\mathbf{x}) = x_3$ is two, and the algebraic order of the singular manifold of the Lorenz system with $h(\mathbf{x}) = x_1$ is one. However, for most applications, we don't know the exact form of the latent dynamical system, and the only practical way is to use the delay-coordinate mapping, which obscures our selection task. The degree of singular manifold can somehow reflect the extent of a variable' observability. Theoretically, the higher the order of singular manifold $\mathcal{M}_s$, the lower the observability.

Here, we propose the variable selection algorithm to select the best measurement for embedding. The basic idea is to quantify the complexity of the geometry in each local neighborhood centered at each data point. If the dynamics are well-reconstructed in such a neighborhood, the local geometry is quite simple, which means that the leading singular value $\sigma_1$ of a Hankel matrix resulting from the delay-coordinate mapping holds a relatively large percentage. The details of our variable selection algorithm is shown in Appendix B. We evaluate the performance of our method on several benchmark dynamic systems, and the results are shown in Table 1. Our algorithm can select the reconstruction with the best observability. However, other factors like the symmetry properties of the dynamical system, i.e, the $x_3$-projection of Lorenz system also influence the final result (Duan et al. (2025) ).

## 4.2 THE D-CHOPT ALGORITHM

We now propose our method D-CHOPT, which leverage the ODE discovery framework together with the invertible residual networks (iResFlow) Behrmann et al. (2019), an invertible neural network, to learn the diffeomorphism between the Hankel embedded matrix and original state space from the optimal partial observed trajectories selected from the Algorithm 1. Moreover, we add additional losses by taking derivative operation based on the automatic differentiation operator in Pytorch. Our method relies on the approximation ability of Neural Networks to approximate the local diffeomorphism of nonlinear dynamic system on the part of domain $\mathcal{M}_o = \mathcal{M} - \mathcal{M}_s$. The goal is to design the loss function to optimize the sparse analytic form, which serves as the closed-form of ODE system. The structure of our Neural Network is shown in the following Figure 2. There are four types of loss in our network structure, which is shown as follows:

- Loss of ODE: $\mathcal{L}_{\dot{z}} = \left\| \nabla_{\mathbf{h}}\phi(\mathbf{h})\dot{\mathbf{h}} - \mathbf{\Theta}\left(\phi(\mathbf{h})\right)\mathbf{\Xi} \right\|_2^2$

- Reconstruction loss of derivative: $\mathcal{L}_{\dot{h}} = \left\| \dot{\mathbf{h}} - \nabla_{\mathbf{z}}\psi(\phi(\mathbf{h}))\,\mathbf{\Theta}(\phi(\mathbf{h}))\mathbf{\Xi} \right\|_2^2$

- First component Loss: $\mathcal{L}_{z_1} = \|h_{i_1} - z_{i_1}\|_2^2$

- Topology Loss: $\mathcal{L}_{\text{topo}} = \text{RTDL}(\mathbf{H}, \mathbf{Z})$

- Consistency Loss: $\mathcal{L}_{\text{cons}} = \sum_{j=1}^{n} \left\| h_{ij} - \left( \int_{t_1}^{t_j} \mathbf{\Theta}(\phi(\mathbf{h}_i))\mathbf{\Xi}\, dt \right)_1 \right\|_2^2$

- Sparsity regularization: $\mathcal{L}_{\text{reg}} = \|\Xi\|_1$

where $\mathbf{z}$ is the target ODE system to be learned. Its $j$-th time realization of $i$-th component is $z_{i_j}$. The governing equation of the target system is approximated using $r$ basis functions

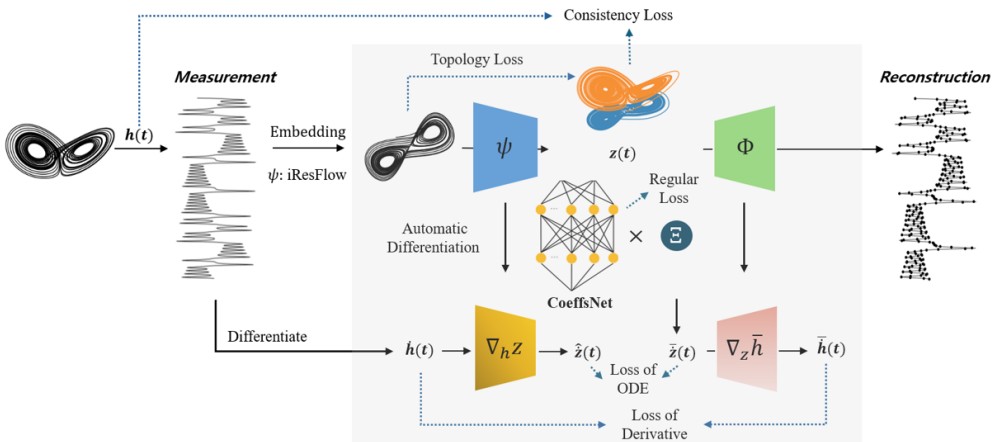

Figure 2: Summary of the network structure of D-CHOPT.

$\theta_1, \theta_2, ..., \theta_r$, say polynomials. $\mathcal{L}_{\dot{z}}$ and $\mathcal{L}_{\dot{h}}$ are loss derived from derivatives for providing physical information for the dynamic system, where $\phi$ represents the mapping from the embedded attractor to the original one while $\psi$ denotes the inversion and we use $\nabla_{\mathbf{h}}\phi(\mathbf{h})\dot{\mathbf{h}} = \dot{\mathbf{z}}$ in the formula. $\Theta(\mathbf{z}) = [\theta_1(\mathbf{z}), \theta_2(\mathbf{z}), ..., \theta_r(\mathbf{z})] \in \mathbb{R}^{m \times r}$ is to form the data matrix, and it works with the coefficient matrix $\Xi = [\xi_1, \xi_2, ..., \xi_r] \in \mathbb{R}^{r \times m}$ for the estimated ODE. $\Xi$ is chosen to be sparse to simplify the system. $\mathcal{L}_{\dot{z}}$ and $\mathcal{L}_{\dot{h}}$ terms play the main role for the ODE estimation. Additionally, to ensure that the attractor manifold of $\mathbf{z}$ preserves topological features of $\mathbf{h}$, we employ the RTD-Lite algorithm (Tulchinskii et al. (2025)) in $\mathcal{L}_{\text{topo}}$, a scalable topological analysis algorithm for manifold matching. The loss term $\mathcal{L}_{z_1}$ requires that the first component of the discovered system equals the observed. Moreover, if the constraint of the first component is satisfied, the integration of the first dimension of time series of $\mathbf{z}$ using the closed-form we learned should follow that of $\mathbf{h}$. The last term $\mathcal{L}_{\text{reg}}$ is designed to make the candidate matrix $\Xi$ sparse. Then the total losses are combined as:

$$\mathcal{L} = \lambda_1 \mathcal{L}_{\dot{z}} + \lambda_2 \mathcal{L}_{\dot{h}} + \lambda_3 \mathcal{L}_{z_1} + \lambda_4 \mathcal{L}_{\text{topo}} + \lambda_5 \mathcal{L}_{\text{cons}} + \lambda_6 \mathcal{L}_{\text{reg}}, \tag{7}$$

where weighting coefficients $\lambda = [\lambda_1, \ldots, \lambda_6]$ are hyperparameters to be tuned.

The proposed structure relies on the invertibility of the fixed point iteration structure of iResFlow and these loss functions constrain the hypothesis class, driving towards a sparse identification. We put the implementation details and the experiment results in Appendix C. Theoretically, if the original dynamic system is $n$-dimensional, it is possible to take the $n$-th derivative into the loss function that aligns with the highest order term of the differential mapping and this network structure can also be transferred to other full-state dynamic system discovery frameworks.

## 5 EXPERIMENT RESULTS

In this section, we demonstrate the ability of the proposed D-CHOPT network to discover governing equations from partially observed trajectories for several canonical dynamic systems. We highlight several key points from our experiments. Figure 3 presents a comparison between SINDy-AE and D-CHOPT. A notable improvement of D-CHOPT is its enhanced ability to discover governing equations by preserving the topological structure.

**Choice of dynamical systems.** In this section, we select four dynamic systems governed by closed-form ODEs, The selected systems exhibit highly nonlinear properties and show complex trajectories. We select multiple ODE systems, including simple linear oscillator, cubic nonlinear oscillator which contains two nonlinearly interacting variables, high-dimensional chaotic Lorenz system and Rossler system which involves three variables forming a strange attractor.

**Measurement and data generating Settings.** For each dynamic system, we sample the ODE system at regular intervals $T = \Delta t, 2\Delta t, ..., n\Delta$. Additionally, we add Gaussian noise (std = 0.01) to the partially observed trajectory of the dynamic system. For each trajectory, we generate each

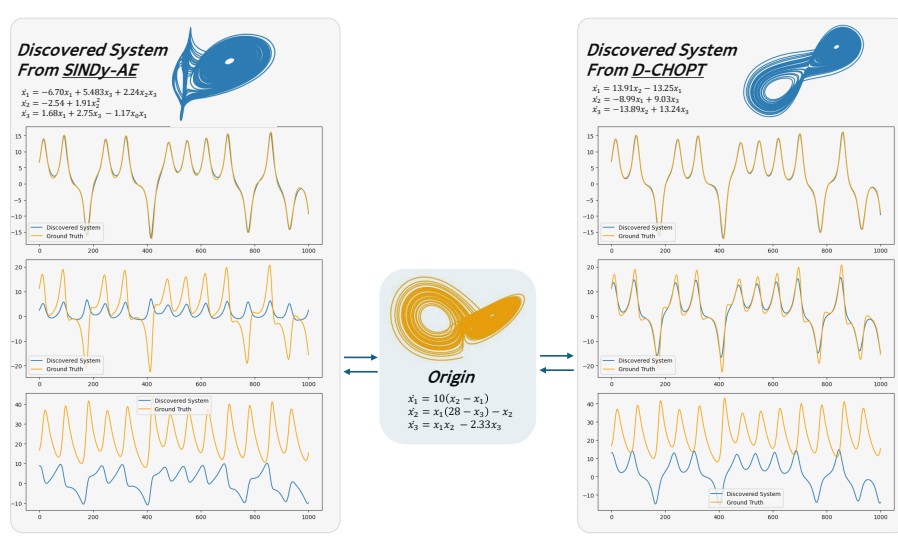

Figure 3: Results for the Lorenz system with measurements given by the coordinate projection of $x_1$. D-CHOPT successfully discovers the closed-form representation while preserving the underlying topological structure.

component of the initial condition $\mathbf{x}_0$ from a uniform distribution of $[-0.5, -0.5]$. Then we obtain the trajectory of the ODE system by solving the initial value problem. The final measurements are contaminated by adding independent Gaussian noise with discrete samples.

**Evaluation Matrices**

In order to make fair comparision, we use three metrics comes from the previous work (Qian et al. (2022)) to evaluate methods. **(1) Success probability (Success Prob.)**: the probability that the **functional form** is correctly recovered from partially observed trajectories. **(2) Distance (Dist.)** between the ground truth $\mathbf{f}^*$ and the learned system $\mathbf{f}$, defined as $d_{\mathbf{x}}(\mathbf{f}, \mathbf{f}^*) := \|\mathbf{f} \circ \mathbf{x} - \mathbf{f}^* \circ \mathbf{x}\|_2 = \|(\mathbf{f} - \mathbf{f}^*) \circ \mathbf{x}\|_2$, where $\circ$ denotes the composition operation and $\mathbf{x}$ represents the noisy full-state measurements obtained from the original system. Note that an incorrect functional term can still yield a small distance. **(3) Sparsity:** the difference in the number of functional terms relative to the ground-truth model (e.g., +1 indicates one extra term; -1 indicates one missing term).

Although many closed-form discovery methods exist, there is only one established framework for partially observed trajectories: the SINDy-Autoencoder (SINDy-AE). It is worth noting that while several other methods can also be applied to discover closed-form systems under incomplete data scenarios, they are all based on SINDy-AE structure. Additional implementation details for the following numerical results are provided in Appendix C.

5.1 NUMERICAL RESULTS

We make fair comparison between the SINDy-AE and our D-CHOPT method based on the following benchmark models.

The linear oscillator is a two-dimensional dissipative ODE system, which is defined as:

$$\dot{x}_1 = \theta_1 x_1 + \theta_2 x_2 \quad \dot{x}_2 = \theta_3 x_1 + \theta_4 x_2, \tag{8}$$

where $\theta_1 = -0.1, \theta_2 = 2.0, \theta_3 = -2.0, \theta_4 = -0.1$. It is easy to check that both measurements provide full observability of the original space.

The nonlinear oscillator consists of two ODEs:

$$\dot{x}_1 = \theta_1 x_1^3 + \theta_2 x_2^3 \quad \dot{x}_2 = \theta_3 x_1^3 + \theta_4 x_2^3, \tag{9}$$

where $\theta_1 = -0.1, \theta_2 = 2.0, \theta_3 = -2.0, \theta_4 = -0.1$. Compared with the linear one, the singular manifold generated by both two measured variables is order-two, which brings difficulties to the discovery of the dynamic system.

The Lorenz63 system is a common test case for chaotic systems, which is defined as:

$$\dot{x_1}(t) = \theta_1(x_2(t) - x_1(t)); \quad \dot{x_2}(t) = x_1(t)(\theta_2 - x_3(t)) - x_2(t); \quad \dot{x_3}(t) = x_1(t)x_2(t) - \theta_3 x_3(t),$$

where $\theta_1 = 10, \theta_2 = 28, \theta_3 = 8/3$. We show the results in Table 2. D-CHOPT achieves comparable and stable performance on both the linear oscillator and the Lorenz63 system. For the Rössler system and nonlinear oscillator case, however, the success probability is lower than that of the SINDy-AE method, since SINDy-AE employs less sparsity, which increases the success probability. Figure 3 illustrates the reason for the performance gain in the Lorenz63 case. When the dimensionality of the dynamical system increases, the attractor manifold undergoes greater deformation, thus making the preservation of its topological structure increasingly important. With the aid of the topology loss, we are able to reduce the degrees of freedom of the dynamical system while simultaneously narrowing the search space of the final solution, thereby accelerating convergence.

| Equation | Method | Success Prob | Dist | Sparsity |
|---|---|---|---|---|
| **Linear Oscillator** | SINDy-AE | **0.9** (0.2) | 4.53 (4.92) | (+1.3) (0.74) |
| | D-CHOPT | **0.9** (0.12) | **0.78** (2.2e-03) | (+1.5) (0.5) |
| **Nonlinear Oscillator** | SINDy-AE | **0.6** (0.2) | 4.42 (1.25e-03) | (+7.4) (2.15) |
| | D-CHOPT | 0.5 (0.27) | **4.41** (2.50e-02) | (+5.8) (1.94) |
| **Lorenz63 System** | SINDy-AE | 0.33 (0.13) | 9.40e+03 (1e+04) | (-2.33) (2.49) |
| | D-CHOPT | **0.57** (0.00) | **4.77e+03** (4.99e+02) | (-0.6) (0.49) |
| **Rossler System** | SINDy-AE | **0.51** (0.07) | 2.13e+02 (2.88e-02) | (1.4) (3.72) |
| | D-CHOPT | 0.43 (0.00) | **2.12e+02** (6.78e+01) | (-1) (0.00) |

Table 2: Three measures Success Prob, the Dist and Sparsity are reported for the four equations. Standard deviations are shown in the brackets.

# 6 DISCUSSION ON FAILURE MODES AND OPEN CHALLENGES

In this work, we explored how the observability of measured variables influences the discovery of ODE systems and we proposed a variable selection algorithm and learning framework. However, discovering latent ODE systems in its closed form is very challenging, and several factors may lead to the failure of the model and present opportunities for future work.

**Extreme observability for high dimensional systems**

For very high-dimensional systems, for example, the nine-dimensional Lorenz system (Reiterer et al. (1998)), at least six dimensions of information are needed to recover a full-observed space, that is, a combination of variables and their derivatives, which suffers the curse of dimensionality in high-dimensional cases. We give a further discussion in Appendix C.

**Candidate Assumptions** One typical feature in our closed-form discovery algorithms for ODE systems is the sparsity on over-supply of the basis candidate functions, meaning that the learning algorithm automatically selects suitable functional terms from the library of candidate functions that determine terms in the estimated ODE $\hat{\mathbf{f}}$. One inevitable problem is that when the order is high when using polynomials (the highest order of polynoial combination), the number of possible candidates becomes large, increasing the searching space dramatically and increase the numerical burden of the discovering procedure. Another obvious problem is if the mathematical expression of the ODE system is complicated, i.e., containing delay terms or fractal terms that are not covered by the candidate functions, the accuracy of the discovered algorithm is limited.

**Noise and slow sampling**

Our algorithm, even the measurement selection algorithm, may fail under large measurement noise or slow sampling cases since the quality of the reconstructed dynamic system relies heavily on the delay-coordinate embedding map, which works well when we have dense and clean samples.

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

## A  DISCUSSION OF IMPLEMENTATION OF TAKENS' EMBEDDING THEOREM

Here, we provide a detailed discussion of Takens' Embedding Theorem, with particular attention to the selection of embedding parameters such as the time delay and embedding dimension. Before beginning, we make the following assumptionm that for attractive nonlinear dynamical systems, a universal property is that trajectories converge rapidly to the attractor manifold regardless of their initial conditions. Thus, our discussion will focus on the attractor manifold itself and on how Takens' Embedding Theorem can be applied to reconstruct this manifold.

For a time-continuous dynamic system, practical constraints necessitate that measurements are taken at a regular sampling interval $T$. Given this sampling rate, the discrete flow denoted as $\xi_T : M \to M$, can be defined and characterized by the equations $\xi_T(\mathbf{x}_t) = \mathbf{x}_{t+T}$ and $\xi_T^{-1}(\mathbf{x}_t)) = \mathbf{x}_{t-T}$ which associate each point state $\mathbf{x} \in M$ the vector $\xi_T(\mathbf{x})$. The sampling rate $T$ represents an integer multiple of the iteration step for discrete dynamic systems.

Direct observation of the full state of the dynamic system is often infeasible. Instead, we have access to observations via a real-valued measurement function $h : \mathbb{A} \to \mathbb{R}$, producing the signals $\{s_i\}_{i \in \mathbb{N}} = \{h(\mathbf{x}_{i \cdot T})\}_i$, for $i = 0, 1, 2...$, where $\mathbf{x}_0$ is the initial state and $\mathbb{A}$ represents the attractor manifold. A *homeomorphism* between two manifolds $M_1$ and $M_2$ is a continuous bijection $f : M_1 \to M_2$, where its inverse function $f^{-1} : M_2 \to M_1$ is also continuous. Moreover, if the homeomorphism and its inverse are smooth, it is a diffeomorphism. An *embedding* is a diffeomorphism from a manifold $M_1$ into another manifold $M_2$, defined as $f : M_1 \to f(M_1) \subset M_2$. An important point is that embeddings are always injective and without self-intersections. Moreover, our goal is to find an embedding to reconstruct the attractor $\mathbb{A}$ from the signal $\{s_i\}_i$. Given the certain measurement function $h$, the following theorem forms the theoretical foundation for attractor reconstruction.

**Theorem 6** (Takens). *Takens (2006) Let $M$ be an $n$-dimensional smooth manifold. If $v$ is a vector field on $M$ with flow $\psi_t$ and $h$ is a measurement function on $M$, then for generic choices of $v$ and $h$, the differential mapping $F_{h,m} : M \to \mathbb{R}^m$ of the continuous dynamic system into $\mathbb{R}^m$ is given by:*

$$F_{h,m}(\mathbf{x}) = (h(\mathbf{x}), \frac{d}{dt}\Big|_0 h(\psi_t(\mathbf{x})), ..., \frac{d^{m-1}}{dt^{m-1}}\Big|_0 h(\psi_t(\mathbf{x}))) \tag{10}$$

*which is an embedding when $m = 2n + 1$, where $m$ is the embedding dimension, $\frac{d}{dt}\Big|_0$ means the derivatives are evaluated at $t = 0$ and the flow $\psi$ satisfies*

$$\frac{d}{dt}\Big|_0 \psi_t(\mathbf{x}) = v(\psi_0(\mathbf{x})) \tag{11}$$

*for every time $t \in \mathbb{R}$.*

The above theorem also holds for discrete dynamic systems with a diffeomorphism $\psi$ on a compact $n$-dimensional manifold $M$ and a measurement function $h$, for which the embedding is defined as Equation (12), where the value of the lag value $\tau$ is an integer multiple of the iteration size. The generic in this theory means that the differential mapping $F_{h,m}$ is an open and dense embedding in the set of all mappings under the measurement function $h$ and the flow $\psi_t$. The best way to understand this is regarding this theorem as a generalization of the Weak Whitney Embedding Theorem Whitney (1944).

**Theorem 7** (Weak Whitney Embedding). *Every $n$-dimensional manifold $M$ embeds in $\mathbb{R}^{2n+1}$.*

This theorem states that any manifold $M$ can be embedded in $\mathbb{R}^{2n+1}$ without self-intersections given an arbitrary mapping. Whitney proves that the optimal linear bound for the minimum embedding dimension is $2n$. Takens theorem demonstrates that the differential mapping (10) satisfies this condition, embedding the compact $n$-dimensional manifold $M$ into the reconstructed space $\mathbb{R}^{2n+1}$, even when considering finite discrete samples.

For practical use, discrete versions of the differential mapping (10) are required when working with signals $\{s_i\}_i$ generated by the discrete flow $\xi_T$ with a specific sampling interval $T$. The most common approach is the delay-coordinate mapping $F_{h,\tau,m}(\mathbf{x}_{i\cdot\tau}) : M \to \mathbb{R}^m$, which is defined as:

$$
F_{h,\tau,m}(\mathbf{x}_{i\cdot\tau}) = \begin{bmatrix} h(\mathbf{x}_{i\cdot\tau}) \\ h(\mathbf{x}_{(i-1)\cdot\tau}) \\ \vdots \\ h(\mathbf{x}_{(i-m+1)\cdot\tau}) \end{bmatrix} = \begin{bmatrix} h(\mathbf{x}_{i\cdot\tau}) \\ h(\xi_\tau^{-1}(\mathbf{x}_{i\cdot\tau})) \\ \vdots \\ h(\xi_\tau^{-m+1}(\mathbf{x}_{i\cdot\tau})) \end{bmatrix}
\tag{12}
$$

where the parameter $\tau = k \cdot T$, for $k \in \mathbb{Z}$ is the lag value, and $m$ is the embedding dimension. Theoretically, for minimal time delay $\tau$, a linear combination of coordinates can approximate the derivative such that the delay-coordinate mapping plays the same role as the differential mapping. The well-defined differential mapping is suitable for analytical purposes. This paper explores the properties of shadow manifolds reconstructed through differential mappings while implementing experiments using delay-coordinate mappings.

In practical scenarios, the sampling rate $T$ of signals often cannot be small enough to accurately approximate the differential and higher-order differentials at the given point. However, by selecting an appropriate lag value $\tau$, the delay-coordinate mapping method can obtain the same result in reconstructing the shadow manifold using a sufficiently small $\tau$. Since chaotic dynamic systems consist of highly nonlinear and coupled variables, the signal obtained from the projection function, which serves as the measurement function, has the potential to recover information from other dimensions. The differential mapping method works by separating coupled information and projecting the observed data—via differentiation—in the direction of maximum linear independence, thereby isolating information about variables that are not directly observed. The critical part lies in accurately recovering information from the unknown dimensions using the observed data.

Thus, selecting the lag value $\tau$ plays a critical role in reconstructing the shadow manifold. If the lag value is suitable, the delay-coordinate mapping $\mathbf{F}_{h,\tau,n}(\mathbf{x}(t))$ is equivalent to the differential mapping $\mathbf{F}_{h,n}(\mathbf{x})$ under an affine transformation, and plays as a diffeomorphism between the shadow manifold and the original attractor. However, the selection of the lag value is not only restricted by external factors like the sampling rate $T$ but also its intrinsic properties. For a continuous-time dynamic system with discrete flow, if $\tau$ is too small, the resulting vectors may be highly linear dependent and redundant, leading to a "squeezed" shadow manifold. Conversely, if $\tau$ is excessively large, the new coordinates may become essentially unrelated, causing the shadow manifold to collapse. Based on the above analysis, we can observe that as $\tau$ increases, the shadow manifold undergoes a "stretch-and-fold" process, as depicted in Fig. 4.

For convenience, we omit the sampling interval $T$ for $\tau$ such that the number of $\tau$ shown in this paper refers to the $k$ in the definition, indicating the number of times the sampling interval $T$, for example, $\tau = 5$ means $\tau = 5T$, where T is the sampling rate or the iteration steps for the discrete dynamic system.

Although the selection of lag value for delay-coordinate mapping is an open problem, several works have been done in this field (Tan et al. (2023); Martin et al. (2024)). The most widely-used method to choose the suitable lag value $\tau$ is the mutual information method (Kim et al. (1999)). The basic idea is to calculate the mutual information between the system's observed values at different lag values and the original observed data and then select the first lag value at which the mutual information transitions from decreasing to increasing as the optimal $\tau$ since this represents the lag value that contains sufficient new information while still maintaining some correlation with the original data. This information-based method is theoretically intuitive, but the resulting values often do not correspond to the points at which the shadow manifold is fully stretched before collapsing. Furthermore, in cases where mutual information monotonically decreases with increasing lag value, this method does not work.

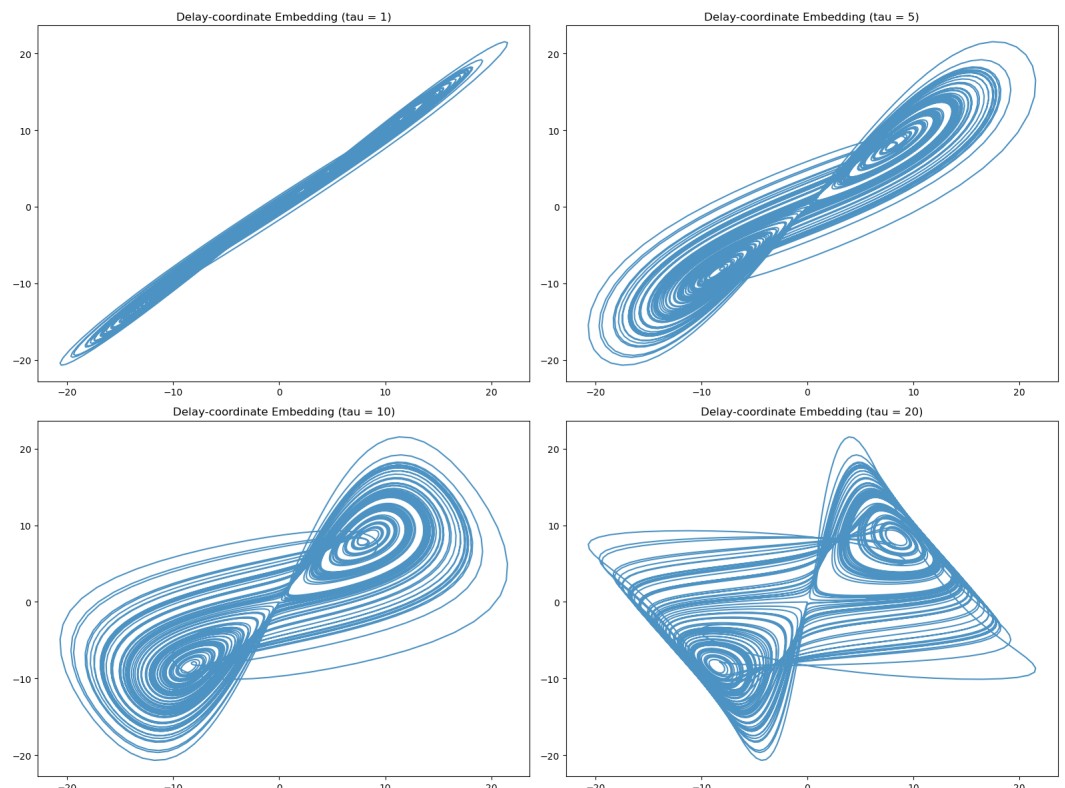

Figure 4: Sequential figures representing the changes in reconstructed shadow manifold $\mathcal{M}_x$ of Lorenz63 system with increasing lag value $\tau$.

According to Takens' theorem, a dynamic system can be embedded in an Euclidean space without self-intersection through any mapping. However, it should be noted that the ideal dimension for embedding an $n$-dimensional dynamic system is $n$. In many cases, the optimal dimension for the shadow manifold generated by differential embedding is also $n$. However, embedding an $n$-dimensional dynamic system into a higher dimension (greater than $n$) does not significantly increase the natural information of the original system, which can be demonstrate using false nearest neighbors Rhodes & Morari (1997), as the redundant information introduced by the extra dimensions does not provide extra information about the original dynamic system and, therefore, does not impact the prediction. Therefore, when the dimension of the original dynamic system is known, the differential embedding method can be directly used to obtain a shadow manifold of the same dimension. In cases where information about the dimensions of the original dynamic system is lacking, false nearest neighbors are a practical approach for dimension selection. In our experiments, we assume that the suitable embedding parameters are selected by brute force selection.

Another important property we need to concern is when the attractor manifold exhibits a symmetric property since the reconstructed attractor manifold may lose the symmetry property of the original attractor. This is because the embedding mods out the symmetry of the attractor manifold, i.e., the attractor manifold reconstructed using the $x_3$ measurement of Lorenz63 system lose the original two-fold rotational symmetry.

## B DETAILS OF OBSERVABILITY AND VARIABLE SELECTION ALGORITHM

Here, we use a concrete example to show the existence of singular manifold $\mathcal{M}_s$ by calculation. We take the Rössler and illustrate why the projection of $x_2$ provides the best observability among all three measurements.

The expression of Rössler system (Rössler (1976)) is:

$$\dot{x} = -y - z$$
$$\dot{y} = x + ay \qquad (13)$$
$$\dot{z} = b + x(z - c)$$

where $a = 0.2, b = 0.2, c = 5.7$. In case when the measurement is $h(\mathbf{x}) = x_1$, the embedding $\Phi_{x_1}$ is:

$$u = x$$
$$v = \dot{x} = -y - z \qquad (14)$$
$$w = \ddot{x} = -x - ay - b - z(x - c).$$

The Jacobian matrix of $\Phi_{x_1}$ is

$$J(\Phi_{x_1}) = \begin{bmatrix} 1 & 0 & 0 \\ 0 & -1 & -1 \\ -1 - z & -a & -(x - c) \end{bmatrix}. \qquad (15)$$

Clearly, the Jacobian matrix $\mathcal{J}(\Phi_{x_1}) = x - c - a$ vanishes for the plane $x = c + a$. That means points located on the plane $x = a + c$ can not be observed from the new coordinate system $(u, v, w)$ through the measurement $h(\mathbf{x}) = x_1$, which is shown in the following figure 5. Although the

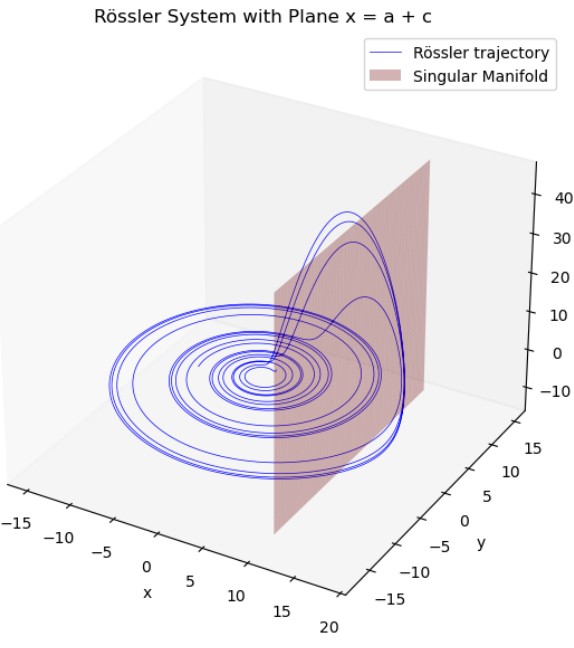

Figure 5: An illustration of Rössler system and the plane $x = a + c$.

Lebesgue measure of the singular manifold which is the intersection part of the Rössler attractor with the plane affects the observability of the attractor but not too much cause it is close to the boundary of the attractor. Similarly, we can calculate the coordinate transformation of projection mapping $h(\mathbf{x}) = x_2$ and $h(\mathbf{x}) = x_3$ using the same procedure. The coordinate transformation $\Phi_{x_2}$ is:

$$J(\Phi_{x_2}) = \begin{bmatrix} 0 & 1 & 0 \\ 1 & a & 0 \\ a & a^2 - 1 & -1 \end{bmatrix}. \qquad (16)$$

The determinant of the Jacobian matrix never vanishes. In other words, $\Phi_{x_2}$ defines a global diffeomorphism between the original attractor manifold and the new attractor manifold. As a result, the $h(\mathbf{x}) = x_2$ provides the best observability of the original system. Similarly, $\Phi_{x_3}$ is:

$$J(\Phi_{x_3}) = \begin{bmatrix} 0 & 0 & 1 \\ z & 0 & (x - c) \\ b + 2z(x - c) & -z & (x - c)^2 - y - 2z \end{bmatrix}. \qquad (17)$$

The determinant of the Jacobian matrix $\Phi_{x_3}$ vanishes for the surface $x^2 = 0$ which is shown in the following figure 6. As a result, the reconstructed attractor manifold $\mathcal{M}_z$ suffers a large shape distortion and collapse near the region of the singular manifold, thus providing the worst observability.

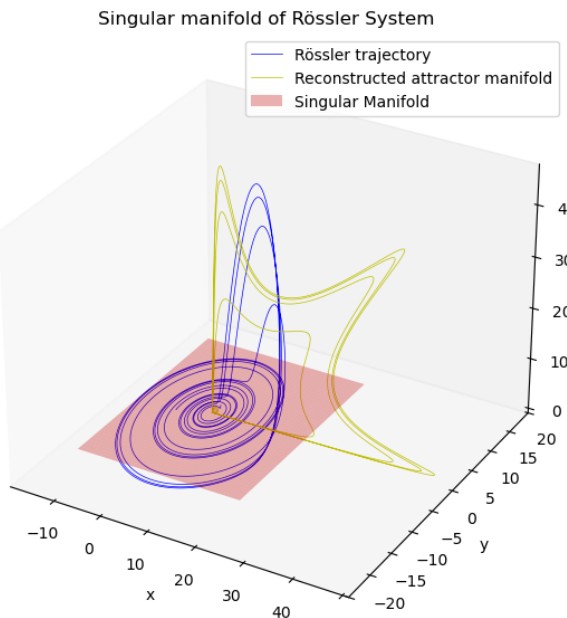

Figure 6: An illustration of Rössler system, reconstructed shadow manifold $\mathcal{M}_z$ and the singular manifold.

The pseudocode of the variable selection algorithm is shown in Algorithm 1.

## C   IMPLEMENTATION DETAILS OF THE EXPERIMENTS

### C.1   DETAILED SETTINGS OF DATASET CURATION

As we discussed in Section 6, the success of the ODE discovery task is highly dependent on the dataset. Instead of only selecting the measurement that provide the best observability through the variable selection algorithm, the measurement settings are also constrained by practical factors such as the sampling rate. The sampling rate directly influences the choice of time delay, for example, if the sampling rate is too low or the data are sampled irregularly, selecting an optimal time delay becomes infeasible. This issue is also an important topic in the time series analysis (Braun et al. (2022)).

Also, the selection of measurement function $h$ is also crucial. In our paper, we restrict the discussion to the case where the measurement function is the coordinate projection of the original system, which preserves the natural properties of the dynamical system. In practice, however, the measurement function could be a linear combination of multiple coordinate projections. D-CHOPT focuses on the discovery task itself, addressing such cases requires domain knowledge of the original system. Nevertheless, incorporating this knowledge can facilitate the closed-form ODE discovery process. Prior knowledge can further enhance the D-CHOPT algorithm, for example, by providing accurate candidate functions or dimensional information and then re-running the D-CHOPT.

---

**Algorithm 1** Hankel-Embedded Local SVD for Observability Assessment

---

1: **Input:** Dataset $\mathcal{D} = \{\mathbf{x}_i(t) \mid i \in \mathbb{N}, t \in [0, T]\}$
2: **Input:** Hankel rows $r$, percentage of center $p$, neighborhood scale $\alpha$, target dimension of dynamic system $td$, number of Monte Carlo samples $M$
3: Calculate the radius per channel: $\epsilon_s \leftarrow \alpha(\max(s) - \min(s))$, $s \in \{x, y, z\}$     ▷ Neighborhood size per channel
4: Calculate the noise scale: $\sigma_s \leftarrow \text{std}(s)$, $s \in \{x, y, z\}$
5: Initialize lists $\mathcal{S}_x, \mathcal{S}_y, \mathcal{S}_z$
6: **for** $m = 1$ to $M$ **do**
7:     Add noise: $\tilde{s} \leftarrow s + \mathcal{N}(0, (0.1\sigma_s)^2)$, $s \in \{x, y, z\}$
8:     **for** $i = 1$ to $r$, $j = 1$ to $(N - r + 1)$ **do**
9:         Construct Hankel matrix: $H_{\tilde{s}}[i, j] \leftarrow \tilde{s}_{i+j}$
10:     **end for**
11:     Perform SVD: $U\Sigma V^\top \leftarrow \text{svd}(H_{\tilde{s}})$
12:     Rank-$td$ projection: $\widehat{C}_{\tilde{s}} \leftarrow V_{1:td}^\top \text{diag}(\Sigma_{1:td})$
13:     Initialize empty list: $\mathcal{A} \leftarrow []$
14:     **for** $j = 1, 1 + p, 1 + 2p, \ldots$ **do**
15:         Define neighbors: $\mathcal{N}_j \leftarrow \{i \mid \|\widehat{C}_{\tilde{s},i} - \widehat{C}_{\tilde{s},j}\|_2 < \epsilon_s\}$
16:         **if** $|\mathcal{N}_j| > \dim(\widehat{C}_{\tilde{s}})$ **then**
17:             Compute centered matrix: $\bar{C} \leftarrow \widehat{C}_{\tilde{s}, \mathcal{N}_j} - \text{rowmean}(\widehat{C}_{\tilde{s}, \mathcal{N}_j})$
18:             Perform SVD: $U\Sigma V^\top \leftarrow \text{svd}(\bar{C})$
19:             Append: $\frac{100\sigma_1}{\sum_i \sigma_i}$ to $\mathcal{A}$ for $i = 1, 2, \ldots, td$
20:         **end if**
21:     **end for**
22:     Compute the mean: $S1_{\text{mean}} \leftarrow \text{mean}(\mathcal{A})$
23:     Append $S1_{\text{mean}}$ to $\mathcal{S}_x, \mathcal{S}_y, \mathcal{S}_z$
24: **end for**
25: **Return:** $(\mathcal{S}_x, \mathcal{S}_y, \mathcal{S}_z)$

---

## C.2 DETAILED SETTINGS FOR EACH EXPERIMENT

The detailed settings for each experiment in Section 5 are shown in Table 3. The time horizon $T$ is the end time point and the initial point is chosen randomly from a given interval. For practical applications, the time horizon and $\Delta t$ are determined by the problem itself.

Table 3: The detailed settings for each simulation: noise level $\sigma_R$, time step size $\Delta t$, total number of trajectories $N$, time horizon $T$, range of initial conditions $[a, b]$.

| System | $\sigma_R$ | $\Delta t$ | $N$ | $T$ | $[a, b]$ |
|---|---|---|---|---|---|
| Linear Oscillator | 0.01 | 0.01 | 2 | 20 | [-0.5, 0.5) |
| Nonlinear Oscillator | 0.01 | 0.01 | 2 | 20 | [-0.5, 0.5) |
| Lorenz System | 0.01 | 0.01 | 5 | 100 | [-0.5, 0.5) |
| Rössler System | 0.01 | 0.01 | 5 | 80 | [-0.5, 0.5) |

## C.3 HYPER-PARAMETER SETTINGS

In our network architecture, we employ iResFlow as both the encoder and decoder, and design an additional sub-network to estimate the parameters of the learned system. Sparsity is enforced through a combination of the regularization loss $\mathcal{L}_{\text{reg}}$ and thresholding. Our strategy is to maintain a mask for sparsity. After a warm-up period, parameters with values below the threshold are pruned, and the learning rates of both iResFlow and the coefficient network are reset. Automatic differentiation is performed using the `torch.func.jacrev`(Paszke et al. (2017)) function, while the consistency loss is computed via the forth-order Runge-Kutta method (Butcher (2007)).

For both the MLP and iResFlow networks, we use a hidden width of 128 and ELU as the activation function (Clevert et al. (2015)). The default MLP architecture is [128, 64, 128] for both encoder and

decoder. In iResFlow, each block is implemented as an iResNetBlock with hidden dimension 128 and residual scaling factor $\alpha = 0.1$, and the total number of iResNetBlock is three.

### C.4 DETAILS OF EVALUATION METRICS

Here, we provide details on how these three metrics are calculated.

**Success Prob**

The success probability is defined as the proportion of correct functional terms successfully identified by D-CHOPT. For example, in the Lorenz63 system, one of our discovered system is:

$$\dot{x}_1(t) = 13.91x_2(t) - 13.25x_1(t); \quad \dot{x}_2(t) = -8.99x_1(t) + 9.03x_2(t); \quad \dot{x}_3(t) = -13.89x_1(t) + 13.24x_3(t),$$

while the original system is:

$$\dot{x}_1(t) = 10(x_2(t) - x_1(t)); \quad \dot{x}_2(t) = x_1(t)(28 - x_3(t)) - x_2(t); \quad \dot{x}_3(t) = x_1(t)x_2(t) - 2.33x_3(t),$$

In this case, four correct terms are recovered: $x_2$ and $x_1$ in $\dot{x}_1$, $x_1$ in $\dot{x}_2$, and $x_3$ in $\dot{x}_3$, yielding a success probability of $4/7$.

**Dist**

To measure how well our learned equation matches the original ones, the definition of vector field discrepancy between a discovered vector field $\mathbf{f}_\theta$ and the ground truth $\mathbf{f}^*$ is:

$$D(\mathbf{f}_\theta, \mathbf{f}^*) = \left( \int_\Gamma \|\mathbf{f}_\theta(x) - \mathbf{f}^*(x)\|_2^2 \, d\mu(x) \right)^{1/2}, \tag{18}$$

where $\Gamma$ refers to the domain of integration (attractor manifold), $\mu$ is the measures over the domain and we take the $L^2$ norm over functions by computing pointwise distance and integrating over the trajectory then taking square root. For practical conditions, we use the empirical version along a sampled trajectory and approximate the integral by a Riemann sum as:

$$D(\mathbf{f}, \mathbf{f}^*) \approx \sqrt{\sum_{i=1}^N \|\mathbf{f}(x_i) - \mathbf{f}^*(x_i)\|_2^2 \, \Delta t} \tag{19}$$

where $\Delta t$ is the time step between samples and $\{x_i\}$ are sampled states from the trajectory.

Theoretically, this distance serves as a functional to measure how well the learned vector field $\mathbf{f}$ approximates the true vector field $\mathbf{f}^*$; the smaller the value, the better the approximation. If the time horizon $T$ is fixed and the sampling step $\Delta t$ is reduced, $D(f, f^*)$ converges to a constant value, with discretization error on the order of $\mathcal{O}(\Delta t)$. From table 2, the values of **Dist.** for the Rössler and Lorenz systems are significantly larger than those for the linear and nonlinear oscillators. This is because, under a fixed sampling step $\Delta t$, a large time horizon $T$ leads to greater error accumulation, and the state ranges of the Lorenz and Rössler systems are wider, further amplifying the discrepancy. Therefore, comparisons are only meaningful across different methods within the same dynamic system.

**Sparsity**

Sparsity is defined as the difference in the number of discovered terms compared to the number of terms in the original system. For example, if the discovered system is :

$$\dot{x}_1(t) = 13.91x_2(t) - 13.25x_1(t); \quad \dot{x}_2(t) = -8.99x_1(t) + 9.03x_2(t); \quad \dot{x}_3(t) = -13.89x_1(t) + 13.24x_3(t),$$

while the original system is:

$$\dot{x}_1(t) = 10(x_2(t) - x_1(t)); \quad \dot{x}_2(t) = x_1(t)(28 - x_3(t)) - x_2(t); \quad \dot{x}_3(t) = x_1(t)x_2(t) - 2.33x_3(t),$$

then the sparsity is -1, meaning that the discovered system contains one fewer functional term than the original.

