# OpenReview forum: "D-CHOPT: DISCOVERING CLOSED-FORM HIGH-DIMENSIONAL ODEs FROM PARTIAL OBSERVED TRAJECTORIES"
_ICLR.cc/2026/Conference — ICLR 2026 Conference Withdrawn Submission_

### Official Review · Reviewer_1WdS · 2025-10-29

**Soundness:** 2
**Presentation:** 2
**Contribution:** 2
**Rating:** 2
**Confidence:** 3

**Summary:**

The paper proposes a framework to recover the closed-form solution of ODEs from partially observable data. Some theories are introduced and the models are evaluated in several systems.

**Strengths:**

The theory and numerical results seem good for several systems.

**Weaknesses:**

1. The paper needs revision to remove typos in notations and reorganize some sections.
2. The experiments are limited.
3. The methods are not well elaborated.

**Questions:**

1.	The author lacks the review of learning equations with partial observability, e.g.,
a.	Li, Haoran, and Yang Weng. "Physical equation discovery using physics-consistent neural network (PCNN) under incomplete observability." Proceedings of the 27th ACM SIGKDD Conference on Knowledge Discovery & Data Mining. 2021.
b.	Stepaniants, George, et al. "Discovering dynamics and parameters of nonlinear oscillatory and chaotic systems from partial observations." Physical Review Research 6.4 (2024): 043062.
2.	Where is your n in your \phi(t;n,\tau)? y(t) is what coordinate from \boldsymbol{y}(t)? Do you mean each entry of \boldsymbol{y}(t)? The author should carefully revise the paper.
3.	If the construction of the H uses only an entry of \boldsymbol{y}(t), does this mean that each coordinate can be independently recovered? This seems to ignore the intra-correlations between ODE variables.
4.	Section 3 seems to be related work, rather than model and algorithm.
5.	I don’t quite understand why should we select measurements for the embedding. The example h(x)=x1 or h(x)=x2 is unclear to me.
6.	It’s hard to understand why we should introduce 6 losses in Section 4.2.
7.	The experiments are limited. For example, there is no ablation study. No sensitivity analysis for the hyper parameters \lambda1 to \lambda6.

---

### Official Review · Reviewer_Uujn · 2025-10-31

**Soundness:** 2
**Presentation:** 3
**Contribution:** 3
**Rating:** 2
**Confidence:** 3

**Summary:**

This paper tackles a challenging and important problem in the realm of scientific machine learning: recovering closed-form ordinary differential equations (ODEs) when only partial trajectories are observed, particularly in high-dimensional dynamical systems. Traditional methods like SINDy or symbolic regression perform well in low-dimensional, fully observed settings, but struggle when the observation is incomplete or the system is chaotic and high-dimensional. To address this, the authors propose D-CHOPT — a novel, theoretically grounded framework that blends delay-coordinate embedding, observability theory, and invertible neural networks (iResFlow) to reconstruct the full state dynamics and then discover sparse, interpretable ODEs.

**Strengths:**

This paper tackles a genuinely hard and underexplored problem — discovering closed-form high-dimensional ODEs from partially observed trajectories. The formulation of D-CHOPT is theoretically grounded, combining observability theory, Takens’ embedding, and invertible neural networks in a coherent way. The authors provide a rigorous discussion of the mathematical underpinnings, including the role of the singular manifold and differential embedding, which is rare in machine learning papers attempting ODE discovery.

Empirically, the experiments on canonical dynamical systems (Lorenz, Rössler, linear/nonlinear oscillators) convincingly demonstrate that D-CHOPT can recover governing equations under conditions where SINDy-AE fails. The inclusion of topology-preserving losses (RTD-Lite) and the clear analysis of observability are both original and significant contributions.

**Weaknesses:**

1. The exposition could be improved — many sections (especially on Takens’ theorem and observability proofs) are mathematically heavy and could be summarized more intuitively for clarity.
2. The experimental evaluation has no demonstration on real or noisy experimental data (e.g., chaotic physical signals, fluid dynamics, or sensor systems).
3. The comparison baselines are narrow — mostly SINDy-AE.
4. Adding some ablation on the influence of topology loss or invertibility constraints. Because it will be helpful to clarify where performance gains actually come from.

**Questions:**

1. How sensitive is D-CHOPT to the choice of embedding dimension and time delay in Takens’ mapping? Can the method adaptively learn these parameters?
2. Can variable selection algorithm handle cases where the observables are nonlinear combinations, not coordinate projections?
3. The topology loss (RTD-Lite) seems critical. Could the authors provide an ablation or visualization showing how the topology is preserved versus SINDy-AE?
4. Can author do more comparison baselines, not just only SINDy-AE?

---

### Official Review · Reviewer_2aMj · 2025-10-31

**Soundness:** 2
**Presentation:** 3
**Contribution:** 2
**Rating:** 2
**Confidence:** 4

**Summary:**

This paper introduces D-CHOPT which aims to identify analytic ODE equations from limited observations of dynamical systems.
The approach builds on Takens’ embedding theorem, combining a variable-selection algorithm (to choose the most observable coordinate) with an invertible neural-network architecture that reconstructsmappings between the observed trajectories and the latent state space.
The model jointly optimizes several losses to recover a parsimonious closed-form ODE. Experiments on canonical systems are compared against SINDy-AE, showing marginal or inconsistent improvements.

**Strengths:**

The paper tackles an important problem: discovering interpretable dynamical laws from partial and noisy observations, which is central to the broader effort of AI-driven scientific discovery. The integration of observability theory and topological constraints into a neural framework is conceptually appealing and demonstrates a thoughtful attempt to link classical dynamical-systems theory with modern machine-learning methods. The exposition includes substantial theoretical background. The overall structure of the method shows a degree of comprehensiveness and reflects an awareness of the main challenges in ODE discovery. The authors also provide qualitative visualizations and clear figures that help convey the intuition behind the approach.

**Weaknesses:**

Despite these conceptual merits, the paper suffers from several significant limitations. First, the problem formulation is internally inconsistent: although the method is presented as handling partial observability, it assumes access to all possible observables in order to choose the “best” one. This assumption undermines the realism of the setting and makes the proposed solution somewhat artificial. Moreover, the restriction to reconstructing the system from a single selected variable, rather than using all available observed signals jointly, appears sub-optimal and contrary to how partial observability is usually addressed in practice. Second, while the paper repeatedly claims scalability to high-dimensional systems, all experiments are performed on low-dimensional benchmarks. No evidence is provided that D-CHOPT can handle genuinely high-dimensional cases, so the “high-dimensional” claim remains unsubstantiated.
Third, the main methodological contribution is insufficiently rigorous. The algorithm is only briefly described in the main text and deferred largely to the appendix, with no theoretical guarantees or analysis of convergence. Empirically, it appears ineffective: for instance, the observability scores reported in Table 1 for the Lorenz system are nearly identical, suggesting the method fails to distinguish between variables meaningfully. Fourth, the empirical evaluation is inconclusive. The comparisons with SINDy-AE show no consistent advantage, and in several cases D-CHOPT performs worse. Some reported variances of 0.00 are implausible and likely indicate flaws in the experimental protocol or insufficient statistical sampling. Finally, the experimental conditions are overly benign—using only 1 % Gaussian noise and regular sampling—so the results do not demonstrate robustness to more realistic or challenging data conditions.

Overall, while the paper proposes an interesting synthesis of ideas, the novelty is limited relative to prior frameworks such as SINDy-AE, and the empirical evidence does not convincingly support the claimed improvements. The combination of conceptual inconsistencies, weak empirical validation and lack of theoretical clarity limits the paper’s contribution at this stage.

**Questions:**

See weaknesses above

---

### Official Review · Reviewer_wVRd · 2025-11-01

**Soundness:** 1
**Presentation:** 2
**Contribution:** 1
**Rating:** 2
**Confidence:** 4

**Summary:**

This work proposes a variant of the SINDy autoencoder method for discovering symbolic governing equations from partially observed dynamics.

**Strengths:**

The authors make an interesting observation regarding variable selection and the issue of a singular manifold that prevents a globally well-defined embedding using a small number of delay coordinates.

**Weaknesses:**

While this work presents an interesting theoretical setup, the method itself is not particularly novel. Previous work has already used similar approaches to discover governing equations from partial observations, including Bakarji et al (cited in the paper), as well as other work not cited:
1. https://pubs.aip.org/aip/cha/article/32/6/063101/2835714/Model-selection-of-chaotic-systems-from-data-with
2. https://journals.aps.org/prresearch/abstract/10.1103/PhysRevResearch.6.043062
3. https://www.nature.com/articles/s42005-022-00987-z
The authors do not compare against nor discuss these works in the context of their approach.

The proposed method introduces a few new loss functions into the SINDy autoencoder setup, which further complicates hyperparameter tuning. This tuning is not discussed, nor are ablation studies performed, when introducing these new terms.

The results in Table 2 seem to suggest very similar performance with SINDy autoencoder, calling into question the benefit of this approach. This is also reflected in Figure 3, where the SINDy-AE seems to discover a nonlinear equation that has the right behavior for the Lorenz attractor, whereas the proposed D-CHOPT discovers a purely linear governing equation (according to the equation in the figure) that is therefore not even chaotic. The plots in Figure 3 for D-CHOPT also do not seem to reflect the discovered governing equation, calling into question the validity of the result.

Finally, the authors claim to be dealing with "high-dimensional ODEs", but none of the examples are beyond 3 variables.

**Questions:**

1. Why restrict yourself to using one observation variable, requiring you to devise a method for choosing one, rather than using all available observations? It seems that the proposed method throws away potentially useful information for reconstructing the dynamics.
2. Figure 2 is generally confusing. What is CoeffsNet?
3. Are the plots in Figure 3 showing full trajectories from the discovered governing equations starting from a single initial condition or is something else being plotted?

---

### Note · Authors · 2025-11-23

I have read and agree with the venue's withdrawal policy on behalf of myself and my co-authors.